# Utilizing the ABC Transporter for Growth Factor Production by *fleQ* Deletion Mutant of *Pseudomonas fluorescens*

**DOI:** 10.3390/biomedicines9060679

**Published:** 2021-06-16

**Authors:** Benedict-Uy Fabia, Joshua Bingwa, Jiyeon Park, Nguyen-Mihn Hieu, Jung-Hoon Ahn

**Affiliations:** 1Department of Biological Sciences, Korea Advanced Institute of Science and Technology (KAIST), Daejeon 34141, Korea; benedictfabia@kaist.ac.kr (B.-U.F.); jbmakhanu10@gmail.com (J.B.); hieuristics@kaist.ac.kr (N.-M.H.); 2Department of Chemistry and Biology, Korea Science Academy of Korea Advanced Institute of Science and Technology, Busan 47162, Korea; jyp131@kaist.ac.kr

**Keywords:** ABC transporter, negatively supercharged protein, growth factor, *Pseudomonas fluorescens*, *fleQ*

## Abstract

*Pseudomonas fluorescens*, a gram-negative bacterium, has been proven to be a capable protein manufacturing factory (PMF). Utilizing its ATP-binding cassette (ABC) transporter, a type I secretion system, *P. fluorescens* has successfully produced recombinant proteins. However, besides the target proteins, *P. fluorescens* also secretes unnecessary background proteins that complicate protein purification and other downstream processes. One of the background proteins produced in large amounts is FliC, a flagellin protein. In this study, the master regulator of flagella gene expression, *fleQ*, was deleted from *P. fluorescens* *Δtp*, a lipase and protease double-deletion mutant, via targeted gene knockout. FleQ directs flagella synthesis, so the new strain, *P. fluorescens* *ΔfleQ*, does not produce flagella-related proteins. This not only simplifies purification but also makes *P. fluorescens ΔfleQ* an eco-friendly expression host because it will not survive outside a controlled environment. Six recombinant growth factors, namely, insulin-like growth factors I and II, beta-nerve growth factor, fibroblast growth factor 1, transforming growth factor beta, and tumor necrosis factor beta, prepared using our supercharging method, were successfully secreted by *P. fluorescens ΔfleQ*. Our findings demonstrate the potential of *P. fluorescens ΔfleQ*, combined with our supercharging process, as a PMF.

## 1. Introduction

*Pseudomonas fluorescens* is a psychrotrophic, gram-negative bacterium that has shown potential as an expression host. It uses a type I secretion system (T1SS) in the form of its ATP-binding cassette (ABC) transporter, TliDEF [1], which is composed of TliD, TliE, and TliF. The three components function as the ABC protein, membrane fusion protein, and outer membrane protein, respectively [2]. T1SS, such as TliDEF, bypass the periplasm, allowing the secretion of proteins with diverse sizes [3]. In addition to that, its simple growth conditions and ability to reach acceptable cell density without optimization make *P. fluorescens* an interesting biological vehicle for recombinant proteins [4]. *P*. *fluorescens* also does not accumulate acetate during growth, so the pH of the culture does not substantially decrease [5]. Furthermore, the capability of *P*. *fluorescens* as a useful host has been demonstrated in various material production, including antibiotics [6], plant growth promoters [7], and epidermal growth factor [8].

To utilize the capacity of *P*. *fluorescens* as a protein factory, strains optimized for production have been developed over the past decade. In particular, a double-deletion mutant of *P*. *fluorescens* called *P*. *fluorescens ΔtliA ΔprtA* (*P*. *fluorescens Δtp*) that no longer produces TliA, which interferes with recombinant protein detection, and PrtA, which hydrolyzes secreted recombinant proteins, was created [1]. The creation of *P*. *fluorescens Δtp* has greatly improved the viability of *P*. *fluorescens* as an expression host. However, our work with *P*. *fluorescens Δtp* has revealed an area of improvement that needs to be addressed to obtain a truly efficient protein factory. In addition to the target protein, *P*. *fluorescens Δtp* secretes several background proteins that are of no use. One particular protein that is produced in large quantities is FliC, a flagellar filament structural protein. Flagella usually function in cell motility, adhesion, and virulence for pathogenic species [9]; however, these cellular functions are unnecessary for expression hosts. We hypothesize that a strain of *P*. *fluorescens* without a flagellum will not only secrete fewer background proteins, but will also produce recombinant proteins at higher concentrations, because more resources will be reallocated away from flagella construction. With this reasoning, we created *P*. *fluorescens ΔfleQ,* a strain of *P*. *fluorescens* that no longer produces the unnecessary flagella, enhancing its capacity as a protein factory.

The genome of *P*. *fluorescens* SIK_W1 has been fully sequenced (NZ_CP031450). Using *P*. *fluorescens Δtp*, the *fleQ* gene was targeted for deletion to create *P*. *fluorescens ΔfleQ*, *P. fluorescens* SIK_W1 strain with three deletions. Synthesis of flagella in *Pseudomonads* requires more than 50 genes, including *fliC*, which codes for the protein that *P*. *fluorescens* secretes in excess [10]. Six dedicated genes regulate the transcription of flagella-related proteins, namely, *fleQ*, *fleS*, *fleR*, *fliA*, *flgM,* and *fleN*. Of these genes, *fleQ* appears to be the master regulator [10,11]. Experiments with *P. aeruginosa*, *P. putida*, and *P. fluorescens* showed that mutation of *fleQ* resulted in bacteria lacking flagella [10,11,12,13,14,15,16,17]. FleQ directs flagella synthesis by activating genes involved in flagellar export, localization and regulation of the flagellar apparatus, structural components of the flagellar basal body, and the motor switch complex genes belonging to the *fleSR* operon [11,18]. For these reasons, *fleQ* was targeted for deletion, with the expectation that its deletion would result in the reduced production of unnecessary competing proteins, which, in turn, can lead to increased target protein expression.

In this study, we focused on growth factors because of their potential applications in medicine, cosmeceuticals, and stem cell differentiation [19,20,21]. We produced six recombinant growth factors using *P*. *fluorescens ΔfleQ* successfully, namely: insulin-like growth factor I (IGFI), insulin-like growth factor II (IGFII), beta-nerve growth factor (βNGF), fibroblast growth factor 1 (FGF1), transforming growth factor beta (TGFβ), and tumor necrosis factor beta (TNFβ). The recombinant growth factors were created by negatively supercharging the wild-type proteins because a lower isoelectric point (pI) has been associated with increased efficiency of LARD3-mediated secretion [22]. Enhancement as an expression host was verified by comparing the secretion activities of *ΔfleQ* and *Δtp*. Last, the viability of our ABC-transporter-mediated protein production system was evaluated by checking for secretion activity in large culture volume.

## 2. Materials and Methods

### 2.1. Bacterial Strains and Growth Conditions

All genetic manipulations were performed using *Escherichia coli* XL1-Blue (Stratagene). The expression of each vector was conducted using *E*. *coli* XL1-Blue, *P*. *fluorescens ΔtliA ΔprtA* (*P*. *fluorescens Δtp*) [1], and *P*. *fluorescens ΔtliA ΔprtA ΔfleQ* (*P*. *fluorescens ΔfleQ*) as the host strain. *E*. *coli* XL1-Blue was cultured in lysogeny broth (LB) medium with 60 μg/mL kanamycin (Km) at 37 °C, whereas *P*. *fluorescens* was cultured in LB, M9, or TB medium with 60 μg/mL Km at 25 °C. We used a modified M9 medium, which consists of an ordinary concentration of M9 salts, 2 mM MgSO_4_, 5% glycerol, trace elements (TE), and 0.1 mM CaCl_2_. TE was diluted from a 100× TE stock solution, which was prepared as follows: 13.4 mM EDTA, 3.1 mM FeCl_3_, 0.62 mM ZnCl_2_, 76 μM CuCl_2_, 42 μM CoCl_2_, 162 μM H_3_BO_3_, and 8.1 μM MnCl_2_. LB agar containing 30 μg/mL Km was used to negatively select deletion mutants, and LB agar containing 10% sucrose was used for positive selection. During conjugation, because *P*. *fluorescens* has an innate resistance to ampicillin, 50 μg/mL carbenicillin was added to distinguish *P*. *fluorescens* cells from *E*. *coli* S17-1 cells [23]. The growth of *E*. *coli* harboring different plasmids was calculated from the measured *A*_600_ values based on logistic cell growth curves, where the initial absorbance of each culture tube was set to be the same value by diluting the overnight culture.

### 2.2. Plasmid Construction

All supercharged genes were synthesized at Bionics (Bionics, South Korea). All genes used in this study had the His_6_ sequence artificially added upstream to allow purification using a Ni-NTA column. The genes were amplified following standard PCR methods and then ligated into the pAD2 expression plasmid, which has a small modification in the *tliD* sequence of pDART [24]. The ligation products were transformed into *E*. *coli* XL1-Blue following the standard heat shock method. In this study, all plasmids used had Km resistance, so 30 μg/mL Km was added to the LB plates and 60 μg/mL Km was added to the liquid cultures. The plates were grown at 37 °C for a day. Colony PCR was performed to screen for colonies containing the proper plasmid. Liquid cultures were then prepared using the proper colonies. If not stated otherwise, 5 mL LB was used for the liquid culture during plasmid construction, and the temperature and RPM were set to 37 °C and 180 rpm, respectively. The plasmids were purified using a QIAprep^®^ Spin Miniprep Kit (QIAGEN, Hilden, Germany).

### 2.3. Identification of FliC Using Mass Spectrometry

Trypsin-digested peptides of the gel-eluted protein band were analyzed using matrix-assisted laser desorption ionization–time of flight (MALDI-TOF) at the Korea Basic Science Institute (Seoul, Korea). The 60 kDa protein band in sodium dodecyl sulfate–polyacrylamide gel electrophoresis (SDS-PAGE) was identified using Coomassie staining and was cut for analysis. The protein was digested in-gel with trypsin. To analyze the protein, tryptic peptides were separated and fractionated on an Akta micro-FPLC system (GE Healthcare, North Richland Hills, TX, USA). The mass spectra for each fraction were obtained using a MALDI-TOF/TOF 5800 instrument (AB Sciex, Thermo Fisher, Grand Island, NY, USA) and identified using ProteinPilot v.4.0.8085 against a genome sequencing database of *P*. *fluorescens* SIK_W1 (NZ_CP031450.1).

### 2.4. P. fluorescens ΔfleQ Construction and Verification

The *fleQ* gene with central 1176 bp deletion (*ΔfleQ*) was generated by SOEing PCR method. Fragment 1 was amplified using primer EcoRI_*ΔfleQ*_F1f and *ΔfleQ*_F1r, and fragment 2 was amplified using *ΔfleQ*_F2f and *ΔfleQ*_F2r_HindIII. The *ΔfleQ* gene was inserted into the suicide vector pK19*mobsacB,* which is replicated in *E*. *coli* but not in *P*. *fluorescens*. The single crossover mutant and final *ΔfleQ* of *P*. *fluorescens* were obtained as previously described [1]. To verify successful construction of the *P*. *fluorescens ΔfleQ*, the genomes of wild-type *P*. *fluorescens* SIK_W1 and *P*. *fluorescens ΔfleQ* were amplified by PCR using primers (Table 1). For the motility test, the cells were inoculated into an LB plate containing 0.3% agar by stabbing colonies with a toothpick and incubating them at 25 °C. For the biofilm assay, *P*. *fluorescens* was cultured for 2 days in an M9 medium. After removing the culture media, it was washed with phosphate-buffered saline (PBS), stained with 0.1% crystal violet for 10 min, and washed with PBS three times [25].

### 2.5. Supercharging the Growth Factors

Using the Bayesian conservation score (a measure of the degree of conservation of a given residue among the homologs of the query protein) and AvNAPSA algorithm, suitable amino acid candidates for mutation were identified. The Bayesian conservation score of each residue was calculated using the ConSurf web server (http://consurf.tau.ac.il/, 15 June 2019) and surface amino acids were determined using the Liu group AvNAPSA algorithm [26,27]. Positively charged residues with a conservation score less than five that were found on the surface of the protein’s 3D structure were then replaced by either aspartic (D) or glutamic acid (E). From the candidate residues the following number of amino acids were mutated per growth factor, IGFI: 5; IGFII: 4; BNGF: 7; FGFI: 8; TGFB: 4; TNFB: 5. The amino acid sequences of wild-type and mutated growth factors are given in Appendix A.

### 2.6. Recombinant Protein Expression and Secretion Analyses

The prepared plasmids were electroporated into two strains of *P*. *fluorescens*, namely, *Δtp* and *ΔfleQ,* and then plated on LB (Km) plates. M9 liquid cultures were then prepared from the colonies. If not stated otherwise, *P*. *fluorescence* plates and liquid cultures were grown at 25 °C for 2 days, and in the case of liquid cultures, at 180 rpm. After reaching the stationary phase, the liquid cultures were harvested. The culture was centrifuged at 12,000 rpm for 5 min. The supernatant samples were prepared by diluting the supernatant in a 5× sample buffer, whereas the cell samples were prepared by resuspending the cell pellet in a 1× sample buffer. Protein expression and secretion were analyzed using SDS-PAGE and Western blotting. Proteins in the supernatant were separated using SDS-PAGE in 10% polyacrylamide gels, according to the method developed by Laemmli [28]. The gels were stained using the sun-gel staining solution (LPS Solution, Daejeon, Korea). For Western blotting, after the proteins were separated in 10% polyacrylamide gels, they were then transferred onto a nitrocellulose membrane (Amersham, UK). Polyclonal anti-LARD3 rabbit immunoglobulin G (rIgG) was used as the primary antibody with a 1:3000 dilution in 5% skim milk and anti-rabbit recombinant goat IgG-peroxidase (anti-rIgG goat IgG-peroxidase) was used as the secondary antibody with a 1:5000 dilution in 5% skim milk. Chemiluminescence agents (Advansta, San Jose, CA, USA) were then used to detect the bands. Western blot images were obtained using an Azure c600 imaging system (Azure, Dublin, CA, USA).

### 2.7. Examining Recombinant Protein Secretion in Flask Culture

First, a 400 mL M9 liquid culture of *P*. *fluorescens ΔfleQ* for each recombinant protein was prepared. Km was added at a concentration of 60 μg/mL, and the vessel used for the liquid cultures was a 1 L flask. The cultures were grown at 25 °C until they reached an optical density (OD) of ≥5, which took approximately 36 h. After reaching the stationary phase, the liquid cultures were harvested. The culture was centrifuged at 3500 rpm for 20 min, and supernatant samples were prepared by mixing it with 5× sample buffer. The recombinant protein in the supernatants was then purified following a His-tag purification protocol [29]. Three milliliters of Ni-NTA resin was used, and the protein samples were eluted using 6 times of 0.5 mL 250 mM imidazole and 4 times of 0.5 mL 1 M imidazole.

## 3. Results

### 3.1. Construction and Verification of P. fluorescens ΔfleQ Mutant

As previously stated, recombinant *P*. *fluorescens Δtp* secreted background proteins in addition to the target recombinant proteins. Figure 1A shows that *P*. *fluorescens Δtp* transformed with a plasmid containing recombinant TliA secreted not only TliA, but also background proteins. The secretion of background proteins by *P*. *fluorescens Δtp* remained consistent across different culture media, as seen in Figure 1B. The main background protein was approximately 60 kDa in size and hydrophobic [24]. We analyzed the background protein using MALDI-TOF and identified it as FliC (flagellin). *P*. *fluorescens ΔfleQ* has the major flagella master gene, *fleQ*, deleted with the intention of creating a strain that does not secrete proteins unnecessary for recombinant protein production (Figure 2). To verify the successful construction of the deletion mutant, *P*. *fluorescens Δtp* and *ΔfleQ* were amplified by PCR using the F-PCR primer set (Figure 1C). In order to verify that the *ΔfleQ* strain does indeed produce no background proteins compared to the *Δtp* strain, SDS-PAGE analysis was conducted on *P*. *fluorescens Δtp* and *ΔfleQ,* which were grown on three media, namely, LB, TB, and M9. In all three media, the *ΔfleQ* strain had no bands, whereas the *Δtp* strain had a distinct band corresponding to FliC and multiple unknown weak bands (Figure 1B). Mutation in *fleQ* results in loss of motility and biofilm formation ability. To examine the loss of FleQ function, we examined the biofilm formation ability of *P*. *fluorescens Δtp* and *ΔfleQ* mutant on a glass tube. The biofilms formed on the surface of the tube were quantified by staining with crystal violet. As shown in Figure 1D, *ΔfleQ* did not produce a biofilm. Also, colonies were stabbed on a plate containing 0.3% agar. As a result, *ΔfleQ* showed no swarming motility (Figure 1E). These results indicated the successful construction of *P*. *fluorescens ΔfleQ*. We also observed that *ΔfleQ* mutant grew faster and to a cell density higher than *Δtp* mutant (Figure 1F,G).

### 3.2. Comparison between Recombinant Protein Secretion of P. fluorescens Δtp and ΔfleQ

To evaluate *P*. *fluorescens ΔfleQ* as a protein manufacturing factory (PMF), we compared its recombinant protein secretion efficiency with that of the *Δtp* strain. Comparison of recombinant protein secretion efficiency was performed by transforming pAD2-GFP(−30) [22] and pAD2-tobacco etch virus (TEV) protease into the *Δtp* and *ΔfleQ* strains and growing them in M9 medium. Western blot analysis showed that the *ΔfleQ* strain secreted both GFP(−30) and TEV in higher amounts relative to the *Δtp* strain (Figure 3). These results indicate that *P*. *fluorescens ΔfleQ* is a better recombinant PMF than *P*. *fluorescens Δtp*.

### 3.3. Expression and Secretion of Recombinant Growth Factors in P. fluorescens ΔfleQ

To further verify that *P*. *fluorescens ΔfleQ* is a viable protein factory, we introduced six growth factors into the bacteria, namely, IGFI, IGFII, βNG, FGF1, TGFβ, and TNFβ. We ligated the wild-type and recombinant versions of these six growth factors into pAD2 and transformed them into the *ΔfleQ* strain to determine whether our process for creating recombinant proteins had successfully induced secretion. The samples were grown in M9 media, and Western blotting was performed to check for expression and secretion of these growth factors in *ΔfleQ*. Of the wild-type growth factors, only IGFI showed a sign of secretion, whereas the others were not secreted at all (Figure 4A). Wild-type βNGF, FGF1, TGFβ, and TNFβ were expressed but not secreted, while IGFII was neither expressed nor secreted (Figure 4A). It was discovered that a protein’s acidic isoelectric point (pI) and net-negative charge are major factors that determine their secretion through the *P*. *fluorescens* ABC transporter, TliDEF, and in particular, lowering the pI improved LARD3-mediated protein secretion [22]. Referring to the Bayesian conservation score of each amino acid, we mutated positively charged amino acids (K, R) that are not highly conserved into negatively charged amino acids (D, E). As a result, five of the six recombinant growth factors were completely secreted (Figure 4B). TNFβ(−) stands out from the other recombinant growth factors, because it is the only one that was not completely secreted, but the result is still a vast improvement compared with wild-type TNFβ, which was not secreted at all. From this, we confirmed that our supercharging process is capable of inducing secretion in previously unsecreted proteins, and *ΔfleQ* is a suitable protein factory for these recombinant proteins. Table 2 presents the concentration of the secreted recombinant proteins, estimated from the band density of SDS-PAGE.

### 3.4. Flask-Volume Preparation of P. fluorescens ΔfleQ

Another hurdle of recombinant protein production is upscaling the production process. To have some idea of whether *P*. *fluorescens ΔfleQ* could produce recombinant proteins prepared by our supercharging process in acceptable amounts, we prepared flask-volume cultures that we purified after the cultures reached an OD of at least 5. The proteins were then purified using a Ni-NTA column. As previously mentioned, the His_6_-tag sequence was added to the genes beforehand, allowing the usage of the His-tag purification process. SDS-PAGE and Western blotting were then performed on the purified protein samples to verify that the proteins were secreted. The six recombinant growth factors were successfully secreted, as indicated using the SDS-PAGE and Western blotting results (Figure 5). As expected, purified TGFβ(−) and TNFβ(−) have a lower concentration than the other recombinant growth factors because their original secretion level was also relatively low (Figure 4B). Moreover, in Figure 5A, bands corresponding to the monomer of the recombinant proteins and those corresponding to dimers were observed. In the SDS-PAGE analysis result (Figure 5B), two bands were observed for a given size, which were estimated to be an oxidized and a reduced monomer.

## 4. Discussion

In this study, we created a deletion mutant using *P*. *fluorescens Δtp*, resulting in *P*. *fluorescens ΔfleQ*, a strain lacking the *tliA*, *prtA*, and *fleQ* genes. The deletion procedure was not detrimental to the mutant, and it retained the characteristics of the parent cell. As expected, the *P*. *fluorescens ΔfleQ* mutant did not secrete unnecessary proteins in significant amounts, unlike *P*. *fluorescens Δtp,* which secreted multiple proteins, including FliC. Moreover, the *ΔfleQ* strain secreted recombinant proteins at higher amounts than the *Δtp* strain, as evidenced by Western blotting after recombinant GFP and TEV were introduced into the bacteria. The *ΔfleQ* strain also grew at a rate faster than the *Δtp* strain in LB, TB, and M9. Futhermore, considering the fact that *ΔfleQ* can grow well in minimal media (M9), our findings indicate that *P*. *fluorescens ΔfleQ* is an effective expression host for recombinant proteins that has potential in large-scale production systems.

*P. fluorescens ΔfleQ* originated from *P. fluorescens* SIK_W1, which belongs to *P. fluorescens* phylogroup, one of eight *P. fluorescens* complex phylogroups [30,31,32]. *P*. *fluorescens* is a psychrotrophic bacteria which is widespread in soil, water, and refrigerated foods [33]. *P*. *fluorescens* is not considered a human or animal pathogen [34], and is not expected to have adverse ecological effects on wildlife [34,35]. The presence of *P*. *fluorescens* in refrigerated foods and on the surface of plants suggests that humans have regularly consumed the bacteria [36]. Using only a simple, defined mineral salt medium supplemented with a carbon and nitrogen source, *P*. *fluorescens* can be cultivated to high densities in bioreactors [37]. Moreover, *P*. *fluorescens* effectively enable the secretion of recombinant proteins [38]. Extracellular secretion simplifies downstream processing, allowing easy and cost-effective protein recovery [38,39]. The safety level, high growth rate, cost-effective growth media, and ability to secrete recombinant proteins make *P*. *fluorescens* an ideal PMF.

To create the strain *P*. *fluorescens ΔfleQ*, the *fleQ* gene was deleted from the genome of *P*. *fluorescens Δtp*. The primary reason for the deletion mutation was that the *Δtp* strain secretes a flagellin protein called FliC in excess. Since *fleQ* is the master regulator of the transcription of flagella-related genes [10,11], the deletion of *fleQ* will stop the production of FliC, making purification and other downstream processes easier. Even though the same result can be achieved by deleting just *fliC* instead of *fleQ,* we hypothesized that a *fleQ* deletion mutant would produce recombinant proteins more effectively than a *fliC* deletion mutant. Common functions of flagella are cell motility, adhesion, and virulence for pathogenic species [9]. However, these functions are unnecessary for an expression host. Moreover, resources that the bacteria previously used to produce flagella will be reallocated, which we hypothesize will improve the recombinant protein expression. Another benefit of an expression host that does not have flagella is that it will pose fewer environmental issues outside the laboratory. Without a flagellum, the chances of survival for a bacterium in the wild are low. The *ΔfleQ* mutant grew faster than the wild-type strain (Figure 1F,G). Not producing flagella saves energy and resources, which improves the overall growth rate of the bacteria. Overall, the *P*. *fluorescens ΔfleQ* strain was expected to be a more efficient and safer expression host than the *Δtp* strain.

At least 11 secretion systems have been found in *P. fluorescens*, i.e., the ABC transporter, Sec, Tat, MscL, Holins, main terminal branch, fimbrial usher porin, autotransporter, two-partner secretion families, Hcp, and VgrG [40,41,42]. However, out of these, only the ABC transporter secretes proteins across the inner and outer membranes [40]. The T1SS ABC transporter of *P*. *fluorescens* facilitated recombinant protein production in this study. ABC transporters are inherently simple compared with other secretion systems. ABC proteins have a characteristic structure consisting of a double set of two basic structural elements, a hydrophobic transmembrane domain, and a cytosolic domain that contains a region involved in ATP-binding [43,44,45]. On the other hand, 12 to 16 proteins are involved in T2SS and T5SS requires 10 Sec proteins and other related proteins for transport only across the inner membrane [46]. Moreover, attaching a C-terminal signal sequence recognized by the ABC transporter allows the secretion of recombinant proteins [2]. Utilizing its ABC transporter, TliDEF, *P*. *fluorescens ΔfleQ* can continuously produce recombinant proteins that can be easily purified from the extracellular media.

Growth factors have widespread applications in medicine. If a cheap and viable mass production system for growth factors is set in place, drug prices will potentially decrease, thereby increasing their accessibility. Six recombinant growth factors, namely, IGFI, IGFII, βNG, FGF1, TGFβ, and TNFβ, were successfully produced in this study. IGFI acts downstream of growth hormone, promoting anabolic processes and tissue growth throughout life [47]. IGFI therapy has shown promise in metabolic harmonization, and IGFI deficiency raises the risk of cardiovascular diseases, type 2 diabetes, and metabolic syndrome [48]. IGFII influences fetal cell division and differentiation, and possibly also metabolic regulation during growth in mammals [49]. Polymorphisms of IGFII have been related to cardiovascular risk factors [50,51,52]. NGF is a neurotrophic factor that promotes the growth and survival of mammalian peripheral sensory and sympathetic nerve cells [52]. Evidence has been found which supports the therapeutic properties of NGF on cutaneous cells, cells of the visual system, and certain diseases of the central nervous system [53]. We only produced the beta-subunit of NGF because NGF is synthesized as a precursor, proNGF, which undergoes post-translational modification to generate the active form of NGF, βNGF [54,55]. FGF1 is involved in an array of physiological processes, including mammalian development, angiogenesis, wound healing, adipogenesis, and neurogenesis [56,57,58,59]. The administration of FGF1 resulted in improved insulin sensitivity, while also inhibiting hepatic lipid accumulation and attenuating increases in plasma glucose and insulin due to a high-fat diet [58]. TGFβ is a pleiotropic polypeptide involved in cell proliferation, migration, and differentiation during embryonic development, and it plays a vital role in tissue homeostasis in adults [60]. TGFβ has been found to be useful in the treatment of wounds with impaired healing, mucositis, fractures, ischemia–reperfusion injuries, and autoimmune disease [61]. TNFβ mediates inflammatory and immune responses and regulates an array of biological processes, including cell proliferation and differentiation, apoptosis, and neurotransmission [62,63]. Various polymorphisms of TNFβ are associated with various autoimmune diseases, such as vitiligo, rheumatoid arthritis, and Crohn disease [63,64]. The production of these six growth factors will allow their use for therapy or clear the path for research regarding their therapeutic properties.

To create the recombinant proteins, we designed negatively supercharged versions of the wild-type growth factors. Supercharging is a mutagenesis strategy in which only residues with side chains protruding to the solvent, without molecular interactions, are mutated. The guiding principle for the design of the recombinant proteins was that a lower pI improves LARD3-mediated protein secretion [22]. However, excessive mutations or mutating amino acids integral to protein shape and structure can lead to a loss of function. To lower the pI below an arbitrarily chosen threshold while minimizing the number of mutations, we focused on the positively charged amino acids. Highly conserved amino acids across homologs were not mutated because they are likely to be essential for maintaining the structure, and by extension, the functionality of the growth factor. We mutated positively charged amino acids that are not highly conserved into negatively charged amino acids with these principles in mind.

## 5. Conclusions

The deletion of *fleQ* resulted in a strain of *P*. *fluorescens* that can reliably produce recombinant proteins prepared using our supercharging method. The *ΔfleQ* strain had a higher production yield than the *Δtp* strain, proving that it is the better PMF. Moreover, the absence of *fleQ* allowed us to conclude that flagella are unnecessary for expression hosts. Expression hosts without a flagellum are favorable because they minimize the risk of recombinant microbes escaping into the environment. From this study, we expect that *P*. *fluorescens ΔfleQ*, paired with our supercharging method, will become more prevalent as a recombinant PMF.

## Figures and Tables

**Figure 1 biomedicines-09-00679-f001:**
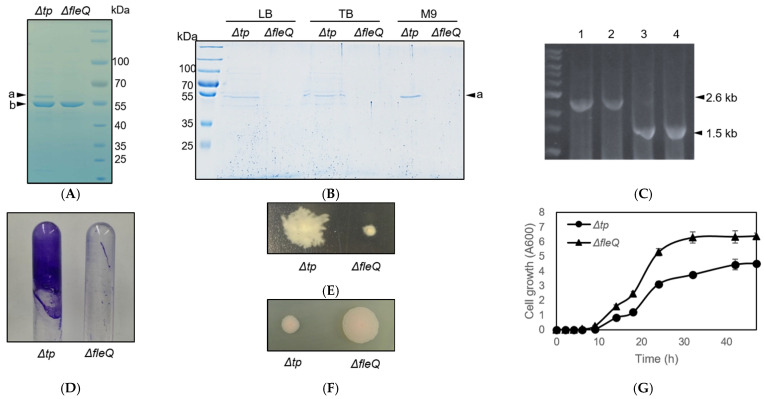
Verification and growth of *P*. *fluorescens ΔfleQ* mutant. (**A**) SDS-PAGE analysis of the proteins released in culture supernatant of *P*. *fluorescens Δtp* and *ΔfleQ* transformed with pAJH10 which secrets TliA. “a“, FliC; “b“, TliA. (**B**) SDS-PAGE analysis of the proteins released in the culture supernatant of *P*. *fluorescens Δtp* and *ΔfleQ*. (**C**) PCR products amplified from four *P*. *fluorescens* chromosomes using the F-PCR primer set. Sample: 1–2; wild type, 3–4; *ΔfleQ*. (**D**) Biofilm formation shown on the glass tube surface by staining with crystal violet. (**E**) Motility of *P*. *fluorescens Δtp* and *ΔfleQ* on LB containing 0.3% agar. The cells were grown at 25 °C for 2 days. (**F**) The colony growth of *P*. *fluorescens Δtp* and *ΔfleQ* on LB. The cells were grown on LB agar plates at 25 °C for 2 days. (**G**) Growth curves of *P*. *fluorescens Δtp* and *ΔfleQ*.

**Figure 2 biomedicines-09-00679-f002:**
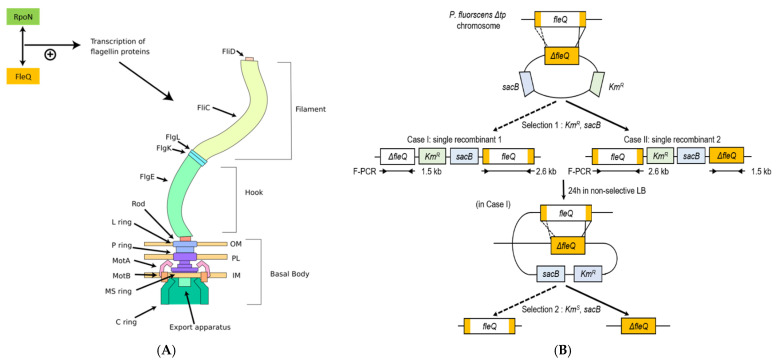
FleQ-regulated flagellin assembly and construction of the *P. fluorescens ΔfleQ* mutant. (**A**) Simplified model depicting the role of FleQ in flagella synthesis and the location of FliC in the flagella structure. FleQ and RpoN positively regulate expression of flagella genes. (**B**) The deletion process to create *P. fluorescens ΔfleQ* and the location of primer-specific sequences used for PCR verifications.

**Figure 3 biomedicines-09-00679-f003:**
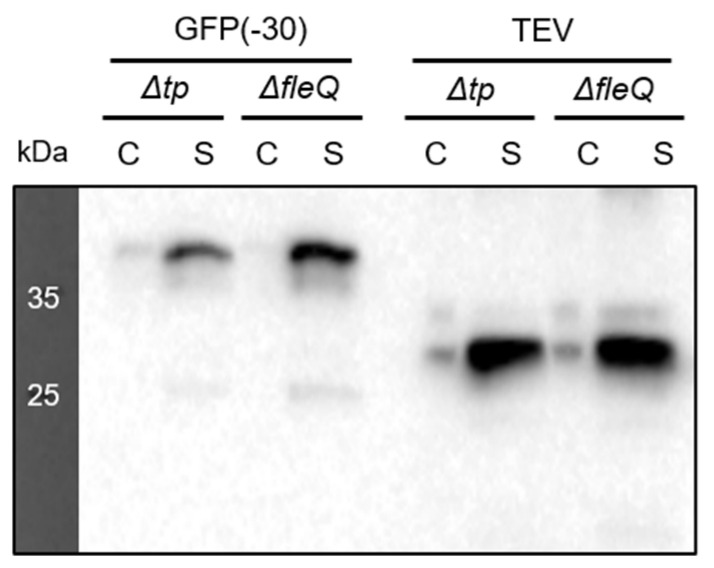
Comparison of *P. fluorescens Δtp* and *ΔfleQ* recombinant protein secretion. The cell lysate is labeled C and the supernatant is labeled S. Intracellular expression and secretion was detected using Anti-LARD3 antibodies. *P. fluorescens* cells were cultured in M9 medium.

**Figure 4 biomedicines-09-00679-f004:**
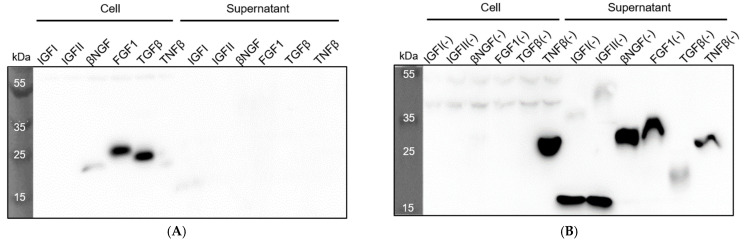
Wild type and recombinant growth factor secretion analysis of *P. fluorescens ΔfleQ*. Wild-type and recombinant proteins were detected using anti-LARD3 antibodies. *P. fluorescens ΔfleQ* cells were cultured in M9 medium for both (**A**) and (**B**). The growth factors were introduced into the cells using the pAD2 plasmid. (**A**) Western blotting of wild-type proteins expressed in cell lysate and secreted culture supernatant by *P. fluorescens ΔfleQ*. (**B**) Western blotting of recombinant proteins expressed in cell lysate and secreted culture supernatant by *P. fluorescens ΔfleQ*.

**Figure 5 biomedicines-09-00679-f005:**
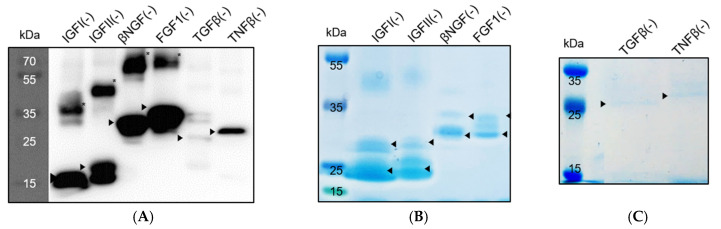
Recombinant growth factor secretion analysis of *P*. *fluorescens ΔfleQ* in flask. (**A**) Western blotting of the purified His-tagged protein from the culture supernatant. (**B**,**C**) SDS-PAGE analysis of the purified His-tagged protein from the culture supernatant. ►, monomer protein; *, dimer protein. The bands which appear in pairs (◄), most visible in (**B**), are the protein’s oxidized and reduced forms of proteins.

**Table 1 biomedicines-09-00679-t001:** List of primers used in this study.

Primer ID	Primer Sequence for PCR
EcoRI_ΔfleQ_F1f	F 5′ GGG GAATTC AGAGCCTGCTCGCGGTTTA 3′
ΔfleQ_F1r	R 5′ TAGTCCTTGA ACGGCTCGATGACAATGAGC 3′
ΔfleQ_F2f	F 5′ ATCGAGCCGT TCAAGGACTACCTCGGCAAC 3′
ΔfleQ_F2r_HindIII	R 5′ GGG AAGCTT TCGGAGACCCGGGCTTCC 3′
F-PCR	F 5′ GCCGACATGATCGACGAAG 3′ R 5′ TCCAGGGAACGGGTGGCG 3′

F—forward primer; R—reverse primer.

**Table 2 biomedicines-09-00679-t002:** The concentration of secreted recombinant protein.

Recombinant Protein	Mw (kDa) *	Concentration (mg/L)
IGFI(−)	20.6	5.6
IGFII(−)	20.2	6.0
βNGF(−)	25.6	8.6
FGF1(−)	28.1	6.6
TGFβ(−)	25.3	1.1
TNFβ(−)	30.7	0.8

* Molecular weight of each recombinant protein was the sum of original molecular weight of growth factors and LARD3 (11.2 kDa).

## Data Availability

The data presented in this study are available on request from the corresponding author. The data are not publicly available due to privacy.

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
