# Peer review of "Utilizing the ABC Transporter for Growth Factor Production by *fleQ* Deletion Mutant of *Pseudomonas fluorescens"

_biomedicines, 2021, doi:10.3390/biomedicines9060679_

Round 1

Reviewer 1 Report

The present work does not contain the information and detailed description of the materials and methods required for the publication in this journal. Moreover, in my opinion some procedures are not correct, In addiction, results provided are not discussed in detail. Thus, I do not recommend to publish it on Biomedicines

Author Response

It is true that the detailed principle and procedure for supercharging is missing in this paper, but that is because we are submitting a paper focusing on supercharging enzymes  and the principle of supercharging in parallel. In this paper we focused more on knockout P. fluorescens and using it for growth factor production. 

To indicate the motivation for this study, we added information about the potential applications of the growth factors (Page 2 line 29). We also specified the number of mutations that each recombinant growth factor has (Page 4 line 4). Lastly, grammatical errors and poorly constructed sentences were corrected to increase the clarity of the paper.

Reviewer 2 Report

The authors have demonstrated that their deletion strain which is a Delta-tp strain that has a further deletion of fleQ is an enhanced biotechnological strain that has a faster growth rate, it secretes unnecessary proteins in smaller amounts and it secretes recombinant proteins at higher amounts.

The paper is well written, easy to follow and with very few grammatical errors.

Some minor corrections:

The Recombinant growth factors needed to have some of their basic amino acids mutated into acidic amino acids. Are these mutated recombinant growth factors of any use for humans?

I did not see some discussion about alternative systems that are already in use for such growth factors and how does this Delta-fleQ strain compare to existing systems.

Page 2, line 14. I think the authors mean that the sequence of this particular deletion strain has been sequenced by them, right? I checked at NCBI and this genome has not been published yet in some journal. I guess that it is part of this paper? From my understanding, this is a P. fluorescens SIK W1 strain with three deletions.

  1. fluorescens taxonomy is very problematic. The name P. fluorescens has been used for many different Pseudomonas strains that do not belong to the same species and they are not monophyletic, rather, they are polyphyletic. In a way, P. fluorescens is a very generic name, referring to a wide lineage (orelse species complex) that includes strains named P. fluorescens and strains with other species names. Where exactly is this Delta-fleQ strain within the P. fluorescens species complex? In my opinion, it would be most useful for the reader to see a phylogenetic or a phylogenomic tree that shows where exactly is this strain in relation to other well-known strains. Please see Nikolaidis et al., 2020 (doi: 10.3390/d12080289) and Garrido-Sanz et al., 2017 (doi: 10.3389/fmicb.2017.00413.) for P. fluorescens taxonomy and discuss this issue in the Discussion section.

Some typos/minor mistakes:

Page 1, line 30 and Page 8, line 25: psychotropic or psychrotrophic?

Page 1, Line 31 and 34. TISS and T1SS. Please use one of them consistently.

Page 3, line 35: Please indicate the exact strain name of the wild type.

Page 3, line 42. Please indicate in this section how many mutations you performed for each of the six growth factors.

Page 5, line 3: “…60 kDa in size and hydrophobic.”

Concerning Figure 2A, did the authors draw it themselves, or did they use some template from KEGG, for example?

Page 7, line 6: “…factors in P. fluorescens…”

Page 9, line 34: 10 sec?

Page 10, line 29: please rephrase.

Is reference 29 correct? See also reference 39.

Author Response

1. The Recombinant growth factors needed to have some of their basic amino acids mutated into acidic amino acids. Are these mutated recombinant growth factors of any use for humans?

We chose the growth factors used in this experiment based on their potential applications in medicine, cosmeceuticals, and stem cell differentiation. This information has been added to the introduction section of the paper (Page 2 line 29). We mutated growth factors for better secretion but we did it with some criteria (Page 3 line44). With the criteria for mutation we listed on the manuscript we also expect that the recombinant growth factors will remain functional and they can be used in the same way as the wild type growth factors (Page 10 line 22-33).

2. I did not see some discussion about alternative systems that are already in use for such growth factors and how does this Delta-fleQ strain compare to existing systems.

We envisioned the paper to be about demonstrating the possibility of ABC transporter-mediated secretion of proteins, which was previously impossible, by using an optimized expression organism and our supercharging method. We do not necessarily claim that this production system is better or worse compared to alternative systems that have already been published.

3. Page 2, line 14. I think the authors mean that the sequence of this particular deletion strain has been sequenced by them, right? I checked at NCBI and this genome has not been published yet in some journal. I guess that it is part of this paper? From my understanding, this is a P. fluorescens SIK W1 strain with three deletions.

You are correct. We have not published the genome sequence in any journal but only in NCBI Genbank. It is cited first in this paper and the strain name is P. fluorescens SIK_W1.  We clarified it in the introduction (Page 2 line 14)

4. fluorescens taxonomy is very problematic. The name P. fluorescens has been used for many different Pseudomonas strains that do not belong to the same species and they are not monophyletic, rather, they are polyphyletic. In a way, P. fluorescens is a very generic name, referring to a wide lineage (orelse species complex) that includes strains named P. fluorescens and strains with other species names. Where exactly is this Delta-fleQ strain within the P. fluorescens species complex? In my opinion, it would be most useful for the reader to see a phylogenetic or a phylogenomic tree that shows where exactly is this strain in relation to other well-known strains. Please see Nikolaidis et al., 2020 (doi: 10.3390/d12080289) and Garrido-Sanz et al., 2017 (doi: 10.3389/fmicb.2017.00413.) for P. fluorescens taxonomy and discuss this issue in the Discussion section.

With regard to a phylogenetic tree showing P. fluorescens SIK W1 strain in relation to other well-known strains, here is a link to one on NCBI that hopefully gives enough context to the strain: https://www.ncbi.nlm.nih.gov/genome/150.

In the above phylogenetic tree, P. fluorescens SIK_W1 is most close to P. fluorescens SBW25 which is one of P. fluorescens type strains. We also analyzed our genome in  type genome server introduced in doi: 10.3389/fmicb.2017.00413 (http://ggdc.dsmz.de/) and found that our strain is Pseudomonas aylmerense and seems to be classified as P. fluorescens phylogroup among eight phylogroups (P. mandelii, P. jessenii, P. koreensis, P. corrugate, P. fluorescens, P. gessardii, P. chlororaphis, P. protegens). We added this content in Discussion (Page 9 line1) and included two additional references (Reference 31, 32).

5. Concerning Figure 2A, did the authors draw it themselves, or did they use some template from KEGG, for example?

We created Figure 2A ourselves. We only wanted to create a simple diagram that gave the readers a clear enough understanding of the function of fleQ.

Some typos/minor mistakes:

Page 1, line 30 and Page 8, line 25: psychotropic or psychrotrophic?

Psychrotrophic is the correct word and the change was reflected on the manuscript (Page 1 line30, Page 9 line3).

Page 1, Line 31 and 34. TISS and T1SS. Please use one of them consistently.

Only T1SS is used in the paper now.

Page 3, line 35: Please indicate the exact strain name of the wild type.

The strain name (SIK_W1) was indicated.

Page 3, line 42. Please indicate in this section how many mutations you performed for each of the six growth factors.

The information was added on Page 4 line 4.

Page 5, line 3: “…60 kDa in size and hydrophobic.”

Page 5, line 3 has been corrected (Page 5 line 6).

Page 7, line 6: “…factors in P. fluorescens…”

Page 7, line 6 has been corrected (Page 7 line 8).

Page 9, line 34: 10 sec?

Sec refers to the general secretory pathway. We rewrote it as “...Sec proteins…” to clarify that we are referring to proteins involved in the general secretory pathway (Page 9 line 40).

Page 10, line 29: please rephrase.

Page 10, line 29 has been edited to be clearer (Page 10 line 37).

Is reference 29 correct? See also reference 39.

Reference 29 has been deleted because book citation format  is not proper (Page 9 line 3). There was two same entities in references so one is deleted (now reference 40). (Page 13 line 6)

All typos and grammatical errors indicated have been corrected.

Reviewer 3 Report

The manuscript by Fabia et al. “Utilizing the ABC transporter for growth factor production by fleQ deletion mutant of Pseudomonas fluorescens” describes the deletion of the master flagellar protein regulator gene fleQ to reduce secretion of endogenous proteins, and enhance the production of recombinant proteins in the media. The authors expressed 6 growth factors in this mutant strain and showed secretion is dependent on eliminating positive charges from the surface to get efficient secretion from the bacteria. They conclude that this strain could be useful for large-scale production of growth factors.

This is an interesting manipulation of a bacterium to enhance recombinant protein production. The increased recombinant secretion (compared to bacteria in which fleQ is not deleted)  likely results from lack of competition with endogenous secreted flagellar proteins. The major problem with this approach, however, is that the wild type growth factors are not secreted, and secretion is only efficient when the positive charges on the surface of the growth factors are mutated to negative charges (that the authors call ‘supercharging’). This is a major drawback for two reasons.  First, there is no demonstration that any of these secreted products retain biological activity when produced with this approach. I think this must be addressed in this paper. I find it surprising that these mutations do not impair folding. Secondly, use of these mutated proteins is likely to induce an immune response in hosts.  A better understanding of the selection mechanism used by the transporter might be worth pursuing. The authors note in the Discussion that previous studies found that a C-terminal peptide recognized by the transporter facilitates secretion of recombinant proteins. It is regrettable that the authors did not engineer this peptide into the wild type growth factors to confer efficient secretion.  By including a protease cleavage site between the secretion peptide and the wild type growth factor, biologically active growth factors could be produced. This recognition peptide could even be used for purification of secreted protein on antibody columns.

Minor issues:

Line 25 reword for clarity-‘the non-production of proteins unnecessary’ perhaps could be ‘reduced production of unnecessary competing proteins’.

Line 39, please include a reference for XL1-Blue.

Author Response

1. The major problem with this approach, however, is that the wild type growth factors are not secreted, and secretion is only efficient when the positive charges on the surface of the growth factors are mutated to negative charges (that the authors call ‘supercharging’). This is a major drawback for two reasons.  First, there is no demonstration that any of these secreted products retain biological activity when produced with this approach. I think this must be addressed in this paper. I find it surprising that these mutations do not impair folding.

We believe the mutated growth factors have biological activity based on the fact that surface charged amino acids interacting with water can be replaced with opposite charged amino acids without change of structure and function (Reference 26, https://doi.org/10.1021/ja071641y). Only varied amino acids were selected as targets of supercharge, which might barely affect the activity of growth factors. As we mentioned in our response to the first reviewer, enzymes supercharged in the same way as in this research retain their enzymatic activities. We are submitting a paper in parallel that further expounds on the supercharging process itself. Regarding the biological activity of the growth factors, it is expected that finding proper cell lines to check the cell proliferation or cell differentiation, on which the signaling pathways also need to be analyzed, would require extensive studies, possibly even requiring an entirely new research project. We are currently planning to do this as a follow-up study.

2. Secondly, use of these mutated proteins is likely to induce an immune response in hosts.  A better understanding of the selection mechanism used by the transporter might be worth pursuing. The authors note in the Discussion that previous studies found that a C-terminal peptide recognized by the transporter facilitates secretion of recombinant proteins. It is regrettable that the authors did not engineer this peptide into the wild type growth factors to confer efficient secretion.  By including a protease cleavage site between the secretion peptide and the wild type growth factor, biologically active growth factors could be produced. This recognition peptide could even be used for purification of secreted protein on antibody columns.

You are correct that mutated growth factors can induce an immune response in humans. However, mutating the growth factors was necessary to secrete them. As we mentioned in our response to your first comment, we applied minimized supercharging so as to keep the recombinant growth factors biologically active and lessen the possibility of inducing an immune response. We intend to explore whether or not the recombinant growth factors induce an immune response in a future study.

All the growth factors used in this paper, both wild type and recombinant alike, were ligated into the pAD2 plasmid which contained LARD3, the C-terminal peptide recognized by the ABC transporter. So all of the proteins were fused with LARD3, with a Factor Xa cleavage site between LARD3 and the protein sequence, allowing for their cleavage later on (Reference 24,doi: 10.1128/AEM.03514-14 ). In addition, the recognition peptide is used for protein purification (Reference 24).

Minor issues:

Line 25 reword for clarity-‘the non-production of proteins unnecessary’ perhaps could be ‘reduced production of unnecessary competing proteins’.

Page 2, line 25 has been reworded.

Line 39, please include a reference for XL1-Blue.

Instead of reference, we added a developer of  XL1-Blue, which is more common (Page 2 line 42).

Round 2

Reviewer 1 Report

I would like to thank the authors for the effort they made to write such good manuscript.

Reviewer 3 Report

Proof of activity involves comparing the supercharged versions against wild type protein in a cell-based assay. Because one form of a supercharged variant is functional does not show any of these are.